# Association of Shunt Size and Long-Term Clinical Outcomes in Patients with Cryptogenic Ischemic Stroke and Patent Foramen Ovale on Medical Management

**DOI:** 10.3390/jcm12030941

**Published:** 2023-01-25

**Authors:** Isis Claire Z. Y. Lim, Yao Hao Teo, Jun Tao Fang, Yao Neng Teo, Jamie S. Y. Ho, Yong Qin Lee, Xintong Chen, Kathleen Hui-Xin Ong, Aloysius S. T. Leow, Andrew Fu-Wah Ho, Yinghao Lim, Ting Ting Low, Ivandito Kuntjoro, Leonard L. L. Yeo, Ching-Hui Sia, Vijay K. Sharma, Benjamin Y. Q. Tan

**Affiliations:** 1Department of Medicine, Yong Loo Lin School of Medicine, National University of Singapore, Level 10, NUHS Tower Block, 1E Kent Ridge Road, Singapore 119228, Singapore; 2Division of Neurology, Department of Medicine, National University Hospital, Singapore 119074, Singapore; 3Department of Emergency Medicine, Singapore General Hospital, Singapore 169608, Singapore; 4Pre-Hospital & Emergency Research Centre, Duke-National University of Singapore Medical School, Singapore 169857, Singapore; 5Centre for Population Health Research and Implementation, Singhealth Regional Health System, Singapore 168753, Singapore; 6Saw Swee Hock School of Public Health, National University of Singapore, Singapore 117549, Singapore; 7Department of Cardiology, National University Heart Centre, Singapore 119074, Singapore

**Keywords:** shunt size, cryptogenic stroke, patent foramen ovale, PFO closure

## Abstract

Introduction: Patent foramen ovale (PFO) is a potential source of cardiac embolism in cryptogenic ischemic stroke, but it may also be incidental. Right-to-left shunt (RLS) size may predict PFO-related stroke, but results have been controversial. In this cohort study of medically-managed PFO patients with cryptogenic stroke, we aimed to investigate the association of shunt size with recurrent stroke, mortality, newly detected atrial fibrillation (AF), and to identify predictors of recurrent stroke. Methods: Patients with cryptogenic stroke who screened positive for a RLS using a transcranial Doppler bubble study were included. Patients who underwent PFO closure were excluded. Subjects were divided into two groups: small (Spencer Grade 1, 2, or 3; *n* = 135) and large (Spencer Grade 4 or 5; *n* = 99) shunts. The primary outcome was risk of recurrent stroke, and the secondary outcomes were all-cause mortality and newly detected AF. Results: The study cohort included 234 cryptogenic stroke patients with medically-managed PFO. The mean age was 50.5 years, and 31.2% were female. The median period of follow-up was 348 (IQR 147-1096) days. The rate of recurrent ischemic stroke was higher in patients with large shunts than in those with small shunts (8.1% vs. 2.2%, *p* = 0.036). Multivariate analyses revealed that a large shunt was significantly associated with an increased risk of recurrent ischemic stroke [aOR 4.09 (95% CI 1.04–16.0), *p* = 0.043]. Conclusions: In our cohort of cryptogenic stroke patients with medically managed PFOs, those with large shunts were at a higher risk of recurrent stroke events, independently of RoPE score and left atrium diameter.

## 1. Introduction

Cryptogenic strokes account for one-third of ischemic strokes, and are defined as strokes of undetermined cause, more than one competing cause, or with incomplete diagnostic investigation [1]. Cardioembolic sources are likely important causes of cryptogenic strokes, and patent foramen ovale (PFO), an incomplete postnatal fusion of the septum primum and secundum, may be a source of cardioembolism [2]. PFO is present in 25% of the general population, and this increases to 40–56% in patients with cryptogenic stroke aged <55 years [3,4]. However, with its high prevalence in the general population, PFO may be incidental in some cases of cryptogenic stroke [5,6]. The American Academy of Neurology guidelines state that it is difficult to determine with certainty whether a patient’s PFO is the cause of stroke, but PFO closure probably reduces stroke risk in selected patients younger than 60 years with embolic-appearing infarction [7]. This is supported by the recently published CLOSE and REDUCE randomized controlled trials (RCT) in patients under 60 years [8,9]. In contrast, earlier RCTs such as the CLOSURE I and PC trial did not show a convincing benefit with PFO closure [10,11]. Furthermore, PFO closure is associated with risks such as atrial fibrillation (AF) and periprocedural complications [8], therefore careful case selection for patients who may benefit from PFO closure is essential.

Characteristics of the PFO, such as right-to-left shunt (RLS), may predispose an individual to ischemic stroke. Theoretically, a greater severity of RLS may increase the risk of paradoxical embolism through the PFO, causing ischemic stroke [12]. In several case-control studies, patients with PFO and acute stroke or transient ischemic attack (TIA) have been reported to have a greater frequency and size of RLS on TEE or transcranial Doppler (TCD) bubble studies [13,14]. The severity of RLS on TCD appeared to correlate with the risk of paradoxical embolism score (RoPE) in one study [15], suggesting it may be predictive of pathogenic PFO in contrast to incidental PFO [16]. However, other studies found no effect of shunt size on recurrent stroke or death [17,18]. Exploratory subgroup analysis of RCTs equally showed mixed and contradictory results [8,9]. Therefore, further evidence is needed to better characterize the impact of shunt size on the risk of ischemic stroke, which may inform the use of shunt size as a criterion for PFO closure in future studies.

In this cohort study of patients with cryptogenic stroke and medically-managed PFO, we aim to investigate the association of shunt size with the risk of recurrent stroke, mortality and newly detected AF, and to identify predictors of recurrent stroke.

## 2. Methods

We performed a retrospective cohort study at a tertiary stroke center. Between January 2014 and December 2020, all consecutive patients with ischemic stroke or TIA who had undergone a non-invasive TCD bubble study with agitated saline were screened for inclusion [19,20]. Patients were included if they were at least 18 years old and had been diagnosed with cryptogenic stroke (stroke of unknown origin) according to the TOAST classification (one cause, more than one cause, no cause after work-up, and no cause with extensive or complete work-up) [21]. Patients with large artery atherosclerosis, lacunar stroke, cardioembolic causes (such as AF) and other definite causes of stroke were excluded. Cryptogenic stroke patients were excluded if they had a negative TCD bubble study or had undergone a PFO closure (Figure 1).

### 2.1. Data Collection

We collected information on baseline demographic parameters, clinical parameters, stroke characteristics, investigations, anti-thrombotic regimes, and post-stroke events. The demographics and clinical variables were selected based on the guidelines of the American Heart Association and American Stroke Association for stroke prevention [22]. A thrombophilia screen was conducted as part of the young stroke work-up. This comprised screening for lupus anticoagulants, anti-cardiolipin IgMs, anti-cardiolipin IgGs, anti-B2-glycoprotein 1 IgMs, anti-B2-glycoprotein 1 IgGs, protein C levels, protein S levels, antithrombin-3, factor V Leiden mutation, prothrombin mutation, homocysteine levels, antinuclear antibodies (ANA), perinuclear anti-neutrophil cytoplasmic antibodies (p-ANCA), antineutrophil cytoplasmic antibodies (c-ANCA), and D-dimer [23,24]. A positive thrombophilia screen was defined by a positive result on 2 counts, at least 6 weeks apart, consistent with the laboratory criteria of Sapporo preliminary classification criteria for antiphospholipid syndrome [25].

Echocardiographic parameters from transthoracic echocardiogram (TTE) and transesophageal echocardiogram (TEE) were chosen based on the association with recurrent stroke events in previous literature, such as the left atrium (LA) diameter and left ventricular ejection fraction [18,26,27,28]. We calculated the RoPE score of all the patients included in our study. The RoPE score predicts the likelihood of pathogenic versus incidental PFO in patients with cryptogenic stroke [16,29]. Factors included in the score are a history of hypertension, diabetes, stroke or TIA, current smoking, cortical infarct on imaging, and age.

### 2.2. TCD Protocol

Non-invasive TCD bubble study with agitated saline was used as an initial screening method for PFO evaluation in patients with ischemic stroke because it has excellent sensitivity and accuracy for PFO detection [30]. The TCD bubble study was performed according to a previously published protocol [31]. Agitated microbubble saline solution was injected through an 18-gauge catheter inserted into the antecubital vein. Continuous TCD monitoring of at least one middle cerebral artery (MCA) was performed for microembolic signals. The Valsalva maneuverer (VM) was performed 4–6 s after the agitated saline injection. Considerable (about 20%) reduction in MCA flow velocities indicates an adequate VM. TCD monitoring is performed for another 16–20 s. Since the RLS grade is affected by the body position during the test, we performed the procedure in supine and sitting positions, and selected the higher value for grading of shunt severity [31].

### 2.3. Grading of Shunt Sizes

Patients who were detected to have a RLS were further divided into two groups of small and large shunts based on the Spencer logarithmic scale (SLS). There are currently two main systems for grading the size of PFO shunts: the International Consensus Criteria (ICC) or the SLS [32]. In the ICC grading, a shunt is considered present and functional in the presence of at least one air microbubble; whereas in the SLS, the embolic tracts are counted and graded against a 6-level logarithmic scale to further stratify the severity of the shunt. While both grading scales have high accuracy in detecting the presence of a shunt, the SLS has a higher positive predictive value in detecting large and functional RLS compared to the ICC (60% vs. 32%), and a lower false-positive rate [32,33]. Therefore, SLS was used to compare shunt sizes in this study. The Spencer grading of 1 (1–10 microbubbles), 2 (11–30 microbubbles), or 3 (31–100 microbubbles) represents a small shunt, while a Spencer grading of 4 (101–300 microbubbles) or 5 (>300 microbubbles) is a large shunt [32].

### 2.4. Clinical Outcomes

The primary outcome of this study was recurrent ischemic stroke, which was diagnosed with neuroimaging. The secondary outcomes were all-cause mortality and new AF detection. New AF was diagnosed if it lasted ≥30 s on prolonged cardiac monitoring or entire duration on the 12-lead electrocardiogram in accordance with the European Society of Cardiology guidelines [34]. Patients determined to require further monitoring were recommended for an implantable loop recorder (ILR).

### 2.5. Statistical Analysis

Data analysis was performed using IBM SPSS version 27 (IBM Corp., Version 27.0. Armonk, NY, USA). Median, first, and third quartiles (Q1, Q3) were reported for non-parametric variables. Means and standard deviations (SD) were reported for continuous variables. Frequencies and percentages were presented for categorical variables. Student’s *t*-test was used to compare continuous parameters, while Fisher’s exact test or chi-squared test was used to compare categorical parameters.

We compared the baseline characteristics, stroke characteristics, imaging results, and risk of primary and secondary outcomes in patients with large versus small shunts on univariate analysis. To identify variables associated with recurrent stroke in patients with PFO and cryptogenic stroke, univariate analysis comparing patients with recurrent stroke and those without were performed. Significant results were pooled into the multivariate logistic regression model, but significant factors found in literature reviews associated with increased risk of recurrent stroke were also included. A multivariable logistic regression model adjusting for RoPE score and LA diameter > 40 mm was performed for the association of large shunts with recurrent stroke. A time-to-event analysis using the Kaplan–Meier method and log-rank test statistic to compare the risk of recurrent ischemic stroke between small shunts and large shunts was performed. The significance level was set at a *p*-value of <0.05.

### 2.6. Statement of Ethics

This study was granted exemption from requiring written informed consent. The study protocol was reviewed and approved by the National Healthcare Group Domain Specific Review Board, approval number 2021/00623.

## 3. Results

A total of 794 patients with ischemic stroke or TIA and TCD were identified in the study period. The final cohort included 234 patients with cryptogenic stroke or TIA and medically managed PFO, comprising a small shunt group of 135 patients (57.7%) and a large shunt group of 99 patients (42.3%) (Figure 1). The patients had a mean age of 50.5 ± 10.9 years, 31.2% (*n* = 73) were female, and the median period of follow-up was 348 days (Q1–Q3: 147–1096) following the index ischemic stroke. All patients with TCD-confirmed PFO underwent a TTE. A TEE was performed for 22.6% (*n* = 53) of patients, and did not detect the presence of a PFO in 26.4% (*n* = 14) of them. The cohort had a median RoPE score of 6 (Q1–Q3 5–7) and an atrial septal aneurysm was detected in 1.0% (*n* = 2) of the cohort.

### 3.1. Large vs. Small Shunt: Baseline Patient Characteristics

There was no significant difference in age, ethnicity, co-morbidities, or presence of thrombophilia between the large and small shunt groups (Table 1). Similarly, stroke characteristics such as NIHSS, bilateral infarcts and infarcts involving the cortex, and TTE features such as PFO visualization, atrial septal aneurysm, LVEF < 50%, and LA diameter > 40 mm were not significantly different. The RoPE score was not found to be associated with shunt size (large shunt: median 5 (Q1–Q3: 4–7), small shunt: median 6 (Q1–Q3: 4–7), *p* = 0.734). ILR implantation was more frequent in patients with small shunts compared to those with large shunt in our study population, 8.1% (*n* = 11) vs. 2.0% (*n* = 2), *p* = 0.043 (Table 1). Antithrombotic regimes were not found to be associated with shunt size (Table 1).

### 3.2. Large vs. Small Shunt: Primary and Secondary Outcomes

In total, 4.7% (*n* = 11) of patients with PFO had a recurrent stroke episode. Recurrent stroke events were higher in patients with large shunts (8.1%, *n* = 8) compared to those with small shunts (2.2%, *n* = 3) (*p* = 0.036) (Table 2). All-cause mortality occurred in 3% (*n* = 4) of the small shunt group and none of the patients in the large shunt group, which was not statistically significant (*p* = 0.140) (Table 2). New AF detection was similar between small and large shunt groups (small shunt: 1.5% [*n* = 2], large shunt: 1.0% [*n* = 1], *p* = 0.751).

### 3.3. Predictors of Recurrent Stroke

Comparing PFO patients with recurrent stroke versus no recurrent stroke, those with recurrent stroke were more likely to have large shunts (recurrent stroke: 72.7% [*n* = 8], no recurrent stroke: 40.8% [*n* = 91], *p* = 0.036) (Table 3). They were also more likely to be Indian and less likely to be Chinese (*p* = 0.006). All other co-morbidities, stroke characteristics, echocardiographic parameters, and RoPE scores showed no significant difference between patients with recurrent stroke and those without. On Kaplan–Meier analysis, the incidence of recurrent stroke was higher in PFO patients with large compared to small shunts (*p* = 0.035). The risk of recurrent stroke was highest in the first 250 days after the index event, with 12% occurring within the first 250 days in the large shunt group (Figure 2).

After adjusting for shunt size, RoPE score, and LA diameter > 40 mm, multivariate analysis showed that the presence of a large shunt was an independent predictor of recurrent stroke (aOR 4.09, 95% CI: 1.04–16.0, *p* = 0.043) (Table 4). The RoPE score and LA diameter > 40 mm was not found to be significantly associated with recurrent stroke episodes, with aORs of 0.869 (95% CI 0.615–1.23, *p* = 0.428) and 0.667 (95% CI 0.134–3.31, *p* = 0.620), respectively.

## 4. Discussion

The main findings of our cohort study of patients with cryptogenic stroke and medically managed PFO were (1) patients with large shunts were significantly more likely to have recurrent stroke than small shunts, (2) the association of large shunts with recurrent stroke was independent of RoPE score and LA size, and (3) baseline characteristics of patients with or without recurrent stroke were similar, except for ethnicity.

The proportion of recurrent ischemic stroke was 4.7% in this cohort study, which increased to 8.1% in patients with large shunt PFOs. The recurrence rates were higher than those observed in medical arms of several RCTs on PFO closure, which reported a yearly incidence of less than 2% per year [35]. However, in most RCTs, such as the CLOSE, REDUCE, RESPECT, CLOSURE I, and PC trials, only patients <60 years were included [8,9,35]. The DEFENSE-PRO trial, which included patients up to 80 years with a mean age of 54 ± 12 years, reported a two-year ischemic stroke rate of 10.5%, similar to our study [36]. As the risk of recurrent stroke increases with age, the higher mean age may contribute to the higher rate of stroke recurrence observed in our study. Our finding that risk of recurrent stroke was higher in patients with large shunt PFOs was consistent with the subgroup analysis of several RCTs on cryptogenic patients aged <60 years. Exploratory analysis in the REDUCE RCT showed a significant reduction in recurrent stroke with PFO closure and antiplatelet therapy versus antiplatelet alone in patients with moderate-to-large shunts, but not those with small shunts [9]. Although the CLOSE RCT did not find any heterogeneity in treatment effect based on shunt size, PFO closure with antiplatelet therapy significantly reduced recurrent stroke compared to antiplatelets alone [8]. Meta-regression of five RCTs showed probable greater benefit of PFO closure in larger shunts, although this was confounded by larger proportion of patients with moderate-to-large shunts compared to antiplatelet agents rather than anticoagulation [37]. The underlying mechanism for this may be that larger shunts facilitate paradoxical embolism through the RLS more easily [12]. PFO may also cause turbulent blood flow and in situ thrombi formation, which may be affected by the PFO morphology and RLS [12]. In contrast, other studies did not show an association of PFO size with stroke risk. A prospective cohort study of 581 cryptogenic stroke by Mas et al. demonstrated that the presence of PFO and size of PFO on TEE bubble study was not associated with increased risk of recurrent stroke [18]. Similarly, Homma et al. on general ischemic stroke patients also found no association of PFO presence and size of PFO on TEE with recurrent stroke or death [17]. TEE, however, is less sensitive than TCD in detecting RLS [38]. Some patients with PFO may have been missed as reported in the studies by Mas et al. and Homma et al. [17,18], and it should be noted that PFO is likely incidental in stroke subtypes such as lacunar or small vessel disease, therefore PFO studies should focus on the cryptogenic stroke population.

The TEE bubble study is considered the gold standard for the diagnosis of PFO due to its better ability to delineate cardiac anatomy [39], and it is often performed after the TCD screening test. A meta-analysis showed that the TEE had a high sensitivity and specificity of 89% and 91%, respectively, for PFO detection when compared to diagnosis by right heart catheterization, surgery, or autopsy [40]. However, in our cohort of patients with TCD-confirmed PFO who underwent TEE, 26.4% of patients did not have TEE-confirmed PFO. Based on previous studies, there is a wide range of TEE diagnostic accuracy, with 7–27% of TEE found to be false negatives and identified as positive on TCD [38]. The higher proportion of false-negatives on TEE suggests that the compliance and quality of TEE in detecting PFO may be variable in our institution, and we postulate that it may be due to the compliance of patients to perform effective Valsalva maneuvers during TEE, particularly when under sedation and with the TEE probe in the esophagus. Compared to TEE, TCD may detect small PFOs missed by TEE and is a more sensitive test in the detection of PFO and RLS with a sensitivity of 95–98% [41,42]. Furthermore, although TCD has high sensitivity and specificity for detecting RLS, it is unable to locate the site of RLS and may include extracardiac shunts such as pulmonary arterial venous malformations not seen on TEE [43]. Hence, this suggests that a combination of the TEE and the TCD is a complementary test and should be considered in the investigation of patients with a high suspicion of paradoxical emboli to achieve higher diagnostic accuracy [44,45]. In our cohort, TEE did not detect the presence of a PFO in 14 out of the 53 patients, which is consistent with previous studies [38]. Based on TCD, we found that large shunts on Spencer grading were predictive of recurrent stroke, corroborating with the benefit of PFO closure seen in exploratory analysis of recent RCTs. A RCT focused on PFO closure in large shunts identified on TCD or TEE is needed to confirm the observed predictive value of shunt size for recurrent PFO-related stroke.

RoPE score was not associated with shunt size or recurrent stroke in our study of medically managed PFO in cryptogenic stroke. The RoPE score, derived from 12 component studies, predicts the likelihood of PFO presence in patients with cryptogenic stroke, and infers from this if the index stroke event is attributable to the PFO [16]. The prevalence of PFO increased from 23% in RoPE score 0–3 to 73% in RoPE score 9 or 10 [16]. An analysis of pooled data from the CLOSURE-I, RESPECT, and PC trials found that the RoPE score anticipated the relative risk reduction of PFO closure versus medical therapy across all levels of the RoPE score (r = 0.95) [46]. However, this study only included trial populations, thus was limited to patients under 60 years-old, and further validation in real-world cohorts for patient selection for PFO closure is needed. The RoPE study also found that patients with a high RoPE score had a lower risk of ischemic stroke despite having a greater attributability to their PFO, and the risk of recurrent stroke remained high even after PFO closure in patients with a low RoPE score in a prospective registry study [47]. This suggests that RoPE may be useful in identifying a subgroup of cryptogenic stroke with likely PFO-related stroke, but as RoPE scores are probabilistic, a low RoPE score cannot rule out a PFO-related stroke with certainty [16,46]. Another limitation of the RoPE score is the exclusion of PFO anatomical features. In our study, baseline characteristics, including those in the RoPE score, were not associated with shunt size or recurrent stroke. Therefore, we propose that shunt size on TCD bubble study may allow further risk stratification for PFO-associated stroke in addition to RoPE score, and increase the accuracy of PFO-related stroke prediction.

LA enlargement is associated with increased risk of recurrent ischemic stroke [48], but the interaction with PFO-associated stroke is unclear. In a retrospective analysis of 1040 patients referred for PFO closure, patients who underwent PFO closure had larger LA diameter than those that were medically managed, which reduced significantly after PFO closure [49]. A LA diameter of ≥43 mm was also predictive of RoPE > 7 independently of RLS, and was also associated with RLS, atrial septum aneurysm, and multiple ischemic lesions on brain MRI, consistent with embolic strokes [49]. LA size measured by LA volume index was higher in patients with PFO and embolic stroke of undetermined source (ESUS), a subgroup of cryptogenic stroke, than controls with PFO in another study [50]. It has been proposed that RLS through a PFO may disrupt blood flow in the LA, leading to LA dysfunction and an AF-like pattern of blood flow, which may lead to clot formation [49]. However, as LA enlargement is an independent risk for recurrent stroke regardless of presence of PFO [48], it is difficult to ascertain the predictive value of LA size on PFO stroke risk. Interestingly, LA enlargement was not associated with recurrent stroke or shunt size in our study, and the discrepancy may be due to the inclusion of medically managed PFO only, resulting in a population with a smaller LA diameter overall. Results of our study would suggest that LA size is a less important factor in PFO-related stroke, but further studies are needed to clarify this complex relationship.

Although the baseline clinical characteristics were similar between patients with large and small shunts, we observed an association between ethnicity and recurrent stroke. Previous studies examining ethnic differences in patients with PFO found that white patients had larger PFO shunts than Black patients, but none researched a multiethnic Asian setting [51]. Our study demonstrated associations between the Indian ethnicity and recurrent stroke but not large shunt size, suggesting that perhaps the strokes were independent of PFO. Other studies found no significant differences in the prevalence of PFO or risk of recurrent stroke [52]. Further studies are needed to characterize any race or ethnic differences in stroke risk associated with PFO and the underlying mechanisms for this. To the best of our knowledge, this is the first study to evaluate PFO shunt size and recurrent stroke risk in a predominantly Asian population. 

### Strengths and Limitations

Several limitations of this study must be acknowledged. Firstly, this study is a single-center retrospective cohort, and may be susceptible to undiscovered confounding factors, even though we found no significant differences in clinically relevant clinical factors between patients with large and small shunts. Secondly, due to the small sample size of 234 patients, there were low event numbers for several clinical outcomes, particularly mortality and new AF detection, therefore they might be underpowered. It is difficult to assess if the AF may have been the cause of the stroke instead of the incidental stroke finding. In addition, the low number of recurrent stroke events increased the probability of overfitting the multivariable regression model, thus limiting the applicability of the results to other datasets. Thirdly, TCD cannot differentiate between PFO and extracardiac causes of RLS. As TEE is invasive, TEE was performed on only 22.6% (*n* = 53) patients in our study, and their important PFO anatomical characteristics were not included in our analysis. This low rate of evaluation in our real-world cohort was in part contributed by the unwillingness of patients to consider PFO closure even if TEE had confirmed a significant PFO. TCD is recognized to be a good screening test for PFO; it has good sensitivity, is non-invasive, and can be performed without sedation. Therefore, it is increasingly used as the first-line investigation in many centers [53], including our institution. Fourthly, our study cohort excluded patients with PFO closure. In our institution, as per the AHA/ASA guidelines, patients with larger shunt sizes or high-risk PFO were offered PFO closure. Hence, this may introduce a selection bias as the 16 patients who underwent PFO closure may have had larger shunt sizes, and there is a bias towards the inclusion of patients with smaller shunts in the medical arm. Our study provides further evidence for the association of shunt size on the risk of recurrent stroke in medically managed PFO in a real-world cohort of multiethnic Asian patients.

## 5. Conclusions

In our cohort of cryptogenic stroke patients with medically managed PFOs, those with large shunts were at a higher risk of recurrent stroke events, independently of RoPE score or LA diameter. RoPE score alone was not significantly associated with recurrent stroke or shunt size. Future studies may consider adopting shunt size on TCD as a feature of high-risk PFO in the case selection for PFO closure in cryptogenic stroke.

## Figures and Tables

**Figure 1 jcm-12-00941-f001:**
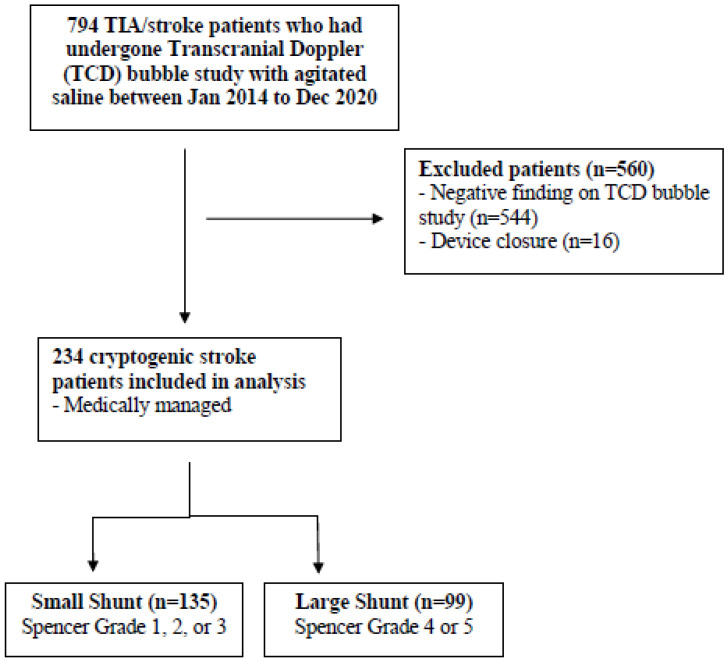
Flowchart showing patient selection, patient inclusion, and exclusion criteria.

**Figure 2 jcm-12-00941-f002:**
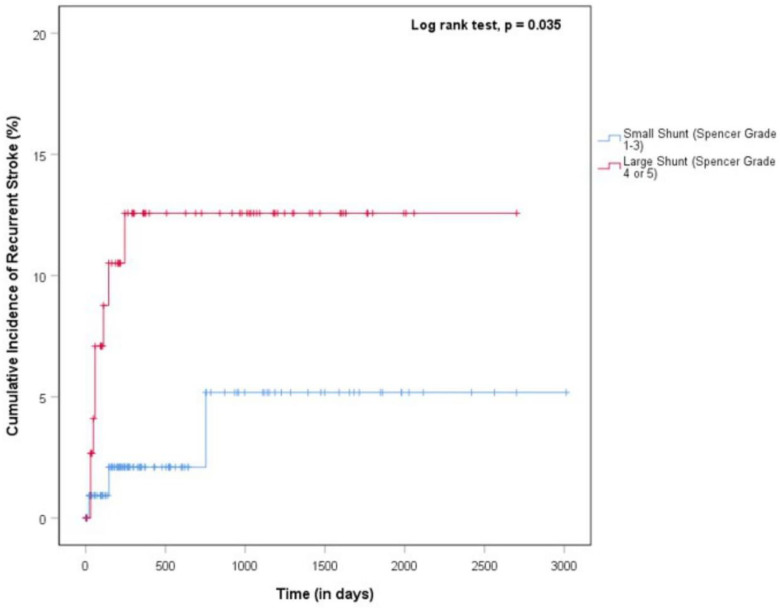
Kaplan–Meier curve comparing cumulative incidence of recurrent stroke between large (Spencer grade 4 or 5) and small (Spencer grade 1, 2, or 3) PFO shunts in patients concomitant cryptogenic stroke.

**Table 1 jcm-12-00941-t001:** Characteristics of PFO patients with small and large shunt.

	Small Shunt (Spencer Grade 1–3) *n* = 135	Large Shunt (Spencer Grade 4–5) *n* = 99	*p*-Value
Patient characteristics			
Age (years) of stroke, mean (SD)	50.2 (12.6)	51.0 (13.9)	0.665
Female, % (*n*)	31.1 (42)	31.3 (31)	0.974
Race, % (*n*)			0.385
Chinese	62.2 (84)	45.5 (45)	
Malay	8.9 (12)	6.1 (6)	
Indian	11.1 (15)	24.2 (24)	
Others	17.8 (24)	24.2 (24)	
Co-morbidities			
Hypertension, % (*n*)	36.3 (49)	28.3 (28)	0.197
Diabetes mellitus, % (*n*)	15.6 (21)	13.1 (13)	0.603
Dyslipidaemia, % (*n*)	19.3 (26)	27.3 (27)	0.148
Ischaemic heart disease, % (*n*)	3.7 (5)	6.1 (6)	0.400
Smoking, % (*n*)	7.4 (10)	14.1 (14)	0.093
Previous stroke/TIA, % (*n*)	10.4 (14)	9.1 (9)	0.745
Chronic kidney disease, % (*n*)	1.5 (2)	1.0 (1)	0.751
Previous venous thromboembolism, % (*n*)	1.5 (2)	1.0 (1)	0.751
Presence of thrombophilia, % (*n*)	13.1 (53/61)	18.5 (10/54)	0.452
Stroke characteristics			
NIHSS on arrival, median (IQR)	6 (1.8–12.5)	8 (0.0–16.3)	0.871
Tissue plasminogen activator, % (*n*)	11.9 (16)	12.1 (12)	0.950
Endovascular thrombectomy, % (*n*)	6.7 (9)	6.1 (6)	0.852
Bilateral infarcts, % (*n*)	10.4 (14)	12.1 (12)	0.674
Infarct involving cortex, % (*n*)	44.4 (60)	54.5 (54)	0.127
Large vessel occlusion, % (*n*)	14.8 (20)	17.1 (17)	0.625
Antithrombotic regime			
Single antiplatelet treatment, % (*n*)	60.7 (82)	52.5 (52)	0.453
Dual antiplatelet treatment, % (*n*)	24.4 (33)	30.3 (30)	0.289
Anticoagulation, % (*n*)			
DOACs	7.4 (10)	6.1 (6)	0.959
Vitamin K antagonist	5.2 (7)	5.1 (5)	1.000
Overall	12.6 (17)	11.1 (11)	0.986
Transthoracic echocardiogram (TTE)			
PFO visualized on TTE, % (*n*)	3.0 (4)	5.1 (5)	0.412
Atrial septal aneurysm, % (*n*)	0.07 (1)	0.1 (1)	0.825
LVEF < 50%, % (*n*)	3.0 (4)	7.1 (7)	0.211
Dilated left atrial size ≥ 40 mm, % (*n*)	19.3 (26)	24.2 (24)	0.420
Others			
Implantable Loop Recorder implantation, % (*n*)	8.1 (11)	2.0 (2)	0.043
RoPE Score, median (IQR)	6 (4–7)	6 (5–7)	0.734

Abbreviations: SD—standard deviation; NIHSS—National Institutes of Health Stroke Scale; IQR—interquartile range; DOACs—dual oral anticoagulant; LVEF—left ventricular ejection fraction; RoPE—risk of paradoxical embolism.

**Table 2 jcm-12-00941-t002:** Long-term outcomes of PFO patients with small and large shunt.

Outcomes	Small Shunt (Spencer Grade 1–3) *n* = 135	Large Shunt (Spencer Grade 4–5) *n* = 99	*p*-Value
No. of days since follow-up, median (IQR)	301 (143–903)	400 (146–872)	0.200
Recurrent ischemic stroke, % (*n*)	2.2 (3)	8.1 (8)	0.036
Mortality, % (*n*)	3.0 (4)	0.0 (0)	0.140
New atrial fibrillation detected, % (*n*)	1.5 (2)	1.0 (1)	0.751

Abbreviations: IQR—interquartile range.

**Table 3 jcm-12-00941-t003:** Univariate analysis evaluating associations of recurrent ischemic stroke.

Variables	No Recurrent Stroke *n* = 223	Recurrent Stroke *n* = 11	*p*-Value
Patient characteristics			
Age (years) of stroke, mean (SD)	50.4 (13.3)	52.3 (11.6)	0.651
Female, % (*n*)	31.4 (70)	27.3 (3)	0.774
Race, % (*n*)			0.006
Chinese	56.1 (125)	36.4 (4)	
Malay	8.1 (18)	0.0 (0)	
Indian	11.1 (33)	54.5 (6)	
Others	21.1 (47)	9.1 (1)	
Large shunt (Spencer grade 4–5), % (*n*)	40.8 (91)	72.7 (8)	0.036
Co-morbidities			
Hypertension, % (*n*)	32.7 (73)	36.4 (4)	0.803
Diabetes mellitus, % (*n*)	14.3 (32)	18.2 (2)	0.725
Dyslipidaemia, % (*n*)	22.4 (50)	27.3 (3)	0.707
Ischaemic heart disease, % (*n*)	4.9 (11)	0.0 (0)	0.451
Smoking, % (*n*)	9.9 (22)	18.2 (2)	0.375
Previous stroke/TIA, % (*n*)	9.9 (22)	9.1 (1)	0.933
Chronic kidney disease, % (*n*)	1.3 (3)	0.0 (0)	1.00
Previous venous thromboembolism, % (*n*)	1.3 (3)	(0)	1.00
Presence of thrombophilia, % (*n*)	16.2 (18/111)	0.0 (0/4)	1.00
Initial stroke characteristics			
Bilateral infarcts, % (*n*)	11.2 (25)	9.1 (1)	0.827
Infarct involving cortex, % (*n*)	49.3 (110)	36.4 (4)	0.401
Large vessel occlusion	15.8 (33)	36.4 (4)	0.056
Imaging studies			
Patent foramen ovale detected on transthoracic echocardiography, % (*n*)	4.6 (9)	0.0 (0)	0.467
Atrial septal aneurysm, % (*n*)	1.0 (2)	0.0 (0)	0.748
LVEF < 50%, % (*n*)	4.0 (9)	18.2 (2)	0.088
Dilated left atrial size ≥ 40 mm, % (*n*)	21.5 (48)	18.2 (2)	0.792
Others			
Implantable Loop Recorder implantation, % (*n*)	5.4 (12)	9.1 (1)	0.600
ROPE score, median (IQR)	6 (5–7)	5 (4–7)	0.448
New atrial fibrillation detected	1.3 (3)	0.0 (0)	0.699

Abbreviations: SD—standard deviation; NIHSS—National Institutes of Health Stroke Scale; IQR—interquartile range; LVEF—left ventricular ejection fraction; RoPE—risk of paradoxical embolism.

**Table 4 jcm-12-00941-t004:** Multivariate logistic regression analysis of factors predicting recurrent stroke.

Variable	Adjusted Odds Ratio (95% CI)	*p*-Value
Large shunt (Spencer grade 4–5)	4.09 (1.04–16.0)	0.043
RoPE score	0.869 (0.615–1.23)	0.428
LA diameter > 40 mm	0.667 (0.134–3.31)	0.620

Abbreviations: CI—confidence Interval; RoPE—risk of paradoxical embolism; LA—left atrial.

## Data Availability

All data generated or analyzed during this study are included in this article. Further enquiries can be directed to the corresponding author.

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
