# Peer review of "Association of Shunt Size and Long-Term Clinical Outcomes in Patients with Cryptogenic Ischemic Stroke and Patent Foramen Ovale on Medical Management"

_jcm, 2023, doi:10.3390/jcm12030941_

Round 1
Reviewer 1 Report
This is a retrospective single-center study aiming to investigate the influence of PFO-related shunt size in stroke recurrence. The study concept is interesting and provides real-world evidence in patients with a particular racial/ethnic background. However, the study has a number of limitations which must be addressed.
In particular, I have the following comments:
Major comments
According to figure 1, out of a population of 250 cryptogenic stroke patients with TCD-detected RLS, 234 were treated medically and 16 underwent PFO closure. The authors should mention the criteria used at their center for PFO closure. Is it likely that patients with large shunts were offered closure instead of medical treatment? If this is the case, then the study population is already biased towards the inclusion of patients with smaller shunts in the medical arm. Nevertheless, since the number of patients who underwent closure is very small, this selection bias is minimized. The authors should include a relevant comment in their limitations section.
The authors mention that TEE was performed in 53/234 (22,6%) patients with TCD confirmed RLS. This is a limitation of the study. Ideally all patients should receive TEE to confirm PFO presence and define high-risk PFO anatomic characteristics such as the presence of ASA, a major determinant of the risk of stroke recurrence. I suppose all patients underwent TTE but this should be clearly stated in the manuscript. TTE may detect ASA but is by far less sensitive than TEE.
Furthermore, TEE did not confirm the presence of PFO in more than a quarter of them (26,4%). This is troublesome. We know that TCD may detect small PFOs missed by TEE (see the extensive review by Koutroulou et al. in Frontiers in Neurology 2020, doi: 10.3389/fneur.2020.00281. eCollection 2020.) but the lack of PFO confirmation in more than 25% of patients exceeds by far what would be expected. A relevant comment should be included in the discussion.
The overall rate of stroke recurrence was 4,7%. Patients with large shunt had recurrence rates at 250 days rising to 12%. This contradicts to the recurrence rates observed in the medical arms of all RCTs where yearly recurrence rates <1-1,5% occurred. The high recurrence rate may imply that another high-risk pathology may have been missed or undisclosed in this particular cohort. True PFO-associated strokes are known to have a low recurrence rate.
RoPE score medians were relatively low in the study cohort regardless of shunt size (6) and counterintuitively were nominally even lower in patients who suffered stroke recurrence (5 vs 6). RoPE score is not the single determinant of stroke recurrence and may be subjected to biases pertaining to the incidence of PFO in the general population with different ethnic/racial background (see the relevant discussion in the paper by Koutroulou I et al. TAND 2020, doi: 10.1177/1756286420964673). The combination of RoPE score to hifh risk anatomic/functional features as proposed by the PASCAL classification may better discriminate pathogenic vs incidental PFOs. The authors should also include (at least a descriptive) analysis of their population for the three degrees of certainty of the PASCAL classification (unlikely, possible, probable).
There is absolutely no mention concerning the antithrombotic regimen the patients were subjected to (single or dual antiplatelet treatment or anticoagulation). As anticoagulation may have an advantage over antiplatelets, the relevant information should be incorporated in the manuscript.
Minor comments
Abstract: Introduction, “……but results have been mixed”. Please replace with “….but results have been controversial”.
“…..new atrial fibrillation, and identify…” Please replace with “….newly detected atrial fibrillation and to identify…”
Conclusion: “…and LA diameter.” Please replace with “….and left atrium diameter.”
Introduction: “In several case-control studies, patients with PFO and acute stroke or transient ischemic attack (TIA) and greater frequency and size of RLS on TEE or transcranial Doppler (TCD) bubble studies.” A verb is missing. Please rephrase.
“The severity of RLS on TCD appeared to correlate with the Risk of Paradoxical Embolism score (RoPE) in one study,[13] suggesting it may be predictive of pathogenic PFO in contrast to incidental PFO, similarly to RoPE based on demographics and co-morbidities.” Please omit the statement after the last comma (similarly…). It does not make sense.
“In this cohort study of patients with cryptogenic stroke and medically-managed PFO, we aim to investigate the association of shunt size with the risk of recurrent stroke, mortality and new AF, and identify predictors of recurrent stroke.” Please replace with ““In this cohort study of patients with cryptogenic stroke and medically-managed PFO, we aim to investigate the association of shunt size with the risk of recurrent stroke, mortality and newly detected AF, and to identify predictors of recurrent stroke.”
Table 3. Please align the comorbidities section so as the numbers coincide with the correct line labels.
Author Response
We thank the reviewer for the insightful comment. We agree that given that our study cohort excluded patients with PFO closure – this may give rise to a selection bias in the study population. Patients with PFO in our institution were offered PFO closure according to the AHA/ASA guidelines, and patients with larger shunt size or high-risk PFO (higher RoPE score such as older age, multiple risk factors) were offered PFO closure. 16 patients underwent PFO closure and were excluded from the cohort.
We have included an additional limitation in the limitation section. For convenient reference, it reads ‘Fourthly, our study cohort excluded patients with PFO closure. In our institution, as per the AHA/ASA guidelines, patients with larger shunt size or high-risk PFO were offered PFO closure. Hence, this may introduce a selection bias as the 16 patients who underwent PFO closure may have had larger shunt size, and there is a bias towards the inclusion of patients with smaller shunts in the medical arm.’ (Page 13 Paragraph 3)
The authors mention that TEE was performed in 53/234 (22,6%) patients with TCD confirmed RLS. This is a limitation of the study. Ideally all patients should receive TEE to confirm PFO presence and define high-risk PFO anatomic characteristics such as the presence of ASA, a major determinant of the risk of stroke recurrence. I suppose all patients underwent TTE but this should be clearly stated in the manuscript. TTE may detect ASA but is by far less sensitive than TEE.
We thank the reviewer for the suggestion. We agree that transoesophageal echocardiogram (TEE) provides the gold standard confirmation for PFO, and is the most sensitive test for identifying PFO. However, owing to limitations in institutional resources, invasiveness of the procedure and patient preference, only 22.6% of patients underwent TEE. All patients underwent transthoracic echocardiogram (TTE) for PFO diagnosis. We have included the latter point into our results, stating ‘All patients with TCD confirmed PFO underwent a TTE. A TEE was performed for 22.6% (n=53) of patients, and did not detect the presence of a PFO in 26.4% (n=14) of them.’ (Page 9 Paragraph 1)
Furthermore, TEE did not confirm the presence of PFO in more than a quarter of them (26,4%). This is troublesome. We know that TCD may detect small PFOs missed by TEE (see the extensive review by Koutroulou et al. in Frontiers in Neurology 2020, doi: 10.3389/fneur.2020.00281. eCollection 2020.) but the lack of PFO confirmation in more than 25% of patients exceeds by far what would be expected. A relevant comment should be included in the discussion.
We thank the reviewer for informing us of this point. We agree with the reviewer that the higher proportion of patients with TCD-identified but TEE-negative PFO is to be noted. In the review by Koutroulou et al., compared to TEE, TCD has a higher pick-up rate and may provide a better diagnostic accuracy of PFO across different age groups – although the proportion of missed diagnosis on TEE (when compared to ICD) is lower. We postulate that the lower pick-up rate for PFO with TEE could be due to the variability in the compliance of TEE. The diagnostic accuracy of PFO on TEE is partly dependent on the ability of patients to perform effective Valsalva manoeuvres, which may be more difficult under sedation, or with the TEE probe in the esophagus. Furthermore, although TCD has high sensitivity and specificity for detecting RLS, it is unable to locate the site of RLS and may include extracardiac shunts such as pulmonary arterial venous malformations not seen on TEE. We thank the reviewer for the highly insightful suggestion and have included this into the discussion, which reads ‘However, in our cohort of patients with TCD-confirmed PFO who underwent TEE, 26.4% of patients did not have TEE-confirmed PFO. Based on previous studies, there is a wide range of TEE diagnostic accuracy, with 7-27% of TEE found to be false negatives and identified as positive on TCD.[34] The higher proportion of false-negatives on TEE suggests that the compliance and quality of TEE in detecting PFO may be variable in our institution, and we postulate that it can be due to the compliance of patients to perform effective Valsalva manoeuvres during TEE, particularly when under sedation and with the TEE probe in the esophagus. Compared to TEE, TCD may detect small PFOs missed by TEE and is a more sensitive test in the detection of PFO and RLS with a sensitivity of 95-98%.[37, 38]. Around 7-27% of TEE were found to be false negatives and identified as positive on TCD.[34] Furthermore, although TCD has high sensitivity and specificity for detecting RLS, it is unable to locate the site of RLS and may include extracardiac shunts such as pulmonary arterial venous malformations not seen on TEE[39]. Hence, this suggests that a combination of the TEE and the TCD are complementary tests and should be considered in the investigation of patients with a high suspicion of paradoxical emboli to achieve higher diagnostic accuracy.[39, 40]’ (Page 11, Paragraph 2)
The overall rate of stroke recurrence was 4,7%. Patients with large shunt had recurrence rates at 250 days rising to 12%. This contradicts to the recurrence rates observed in the medical arms of all RCTs where yearly recurrence rates <1-1,5% occurred. The high recurrence rate may imply that another high-risk pathology may have been missed or undisclosed in this particular cohort. True PFO-associated strokes are known to have a low recurrence rate.
We thank the reviewer for this important comment. While most RCTs showed that the annual probability of stroke was <1-1.5%, it should be noted that these trials, such as the CLOSE, REDUCE, RESPECT, CLOSURE I and PC trial only included patients <60 years. In the DEFENSE-PRO trial which included patients up to 80 years with mean age of 54 ± 12 years reported a 2-year ischemic stroke rate of 10.5%. As the risk of recurrent stroke increases with age, the cohort in our study has a mean age of 50.5±10.9 years, which is higher than most of the previous trials, which may contribute to the higher rate of stroke recurrence in this study. This has been added to the discussion, which reads ‘The proportion of recurrent ischemic stroke was 4.7% in this cohort study, which increased to 8.1% in patients with large shunt PFOs. The recurrence rates were higher than that observed in medical arms of several RCTs on PFO closure, which reported a yearly incidence of less than 2% per year[33]. However, in most RCTs, such as the CLOSE, REDUCE, RESPECT, CLOSURE I and PC trials, only patients <60 years were included[6, 7, 33]. The DEFENSE-PRO trial, which included patients up to 80 years with mean age of 54 ± 12 years, reported a 2-year ischemic stroke rate of 10.5%, similar to our study [34]. As the risk of recurrent stroke increases with age, the higher mean age may contribute to the higher rate of stroke recurrence observed in our study’ (Page 10 Paragraph 3)
RoPE score medians were relatively low in the study cohort regardless of shunt size (6) and counterintuitively were nominally even lower in patients who suffered stroke recurrence (5 vs 6). RoPE score is not the single determinant of stroke recurrence and may be subjected to biases pertaining to the incidence of PFO in the general population with different ethnic/racial background (see the relevant discussion in the paper by Koutroulou I et al. TAND 2020, doi: 10.1177/1756286420964673). The combination of RoPE score to hifh risk anatomic/functional features as proposed by the PASCAL classification may better discriminate pathogenic vs incidental PFOs. The authors should also include (at least a descriptive) analysis of their population for the three degrees of certainty of the PASCAL classification (unlikely, possible, probable).
We thank the reviewer for the constructive comment. We agree that the RoPE score is not the sole determinant of stroke recurrence and may be subject to bias. We also agree that the PASCAL classification is an important tool which incorporates the RoPE Score and functional/anatomic elements of PFO in estimating the probability of causal relationship between PFO and stroke. We sought to assign the degrees of certainty of PFO-related stroke according to the PASCAL classification in our cohort, but this was unfortunately limited by the available data, as the number of bubbles crossing the left atrium was not quantified on TEE scans. However, bubble counting was performed on TCD scans, which allowed us to assign the Spencer grading of right-to-left shunt to each patient. The Spencer grading served as a proxy measure of shunt size approximating the component of “Large shunt size” for the PASCAL classification. The other components of PASCAL, namely the RoPE score and the presence of atrial septal aneurysm, are presented together with the Spencer grade in Table 1. We have added a descriptive analysis of our population according to these components in the Results section as recommended by the reviewer. The sentence reads, ‘The cohort had a median RoPE score of 6 (Q1-Q3 5-7) and an atrial septal aneurysm was detected in 1.0% (n=2) of the cohort.’ We thank the reviewer for the opportunity to improve the quality of our manuscript.
There is absolutely no mention concerning the antithrombotic regimen the patients were subjected to (single or dual antiplatelet treatment or anticoagulation). As anticoagulation may have an advantage over antiplatelets, the relevant information should be incorporated in the manuscript.
We thank the reviewer for the suggestion. We have incorporated the data regarding antithrombotic regime in our cohort, categorized into single antiplatelet, dual antiplatelet, and anticoagulation regimes. We also calculated the two-sample t-test which showed that there is no statistically significance difference observed in the proportion of antithrombotic regimes used between patients with smaller vs larger shunts. This is incorporated into Table 1 and described in results section. For convenient reference, it reads ‘Antithrombotic regimes were not found to be associated with shunt size (Table 1).’ (Page 9 Paragraph 2)
Minor comments
Abstract: Introduction, “……but results have been mixed”. Please replace with “….but results have been controversial”.
“…..new atrial fibrillation, and identify…” Please replace with “….newly detected atrial fibrillation and to identify…”
Conclusion: “…and LA diameter.” Please replace with “….and left atrium diameter.”
Thank you for the feedback. We have amended it as per the suggestion of the reviewer.
Introduction: “In several case-control studies, patients with PFO and acute stroke or transient ischemic attack (TIA) and greater frequency and size of RLS on TEE or transcranial Doppler (TCD) bubble studies.” A verb is missing. Please rephrase.
Thank you for pointing this out. We have rephrased as ‘In several case-control studies, patients with PFO and acute stroke or transient ischemic attack (TIA) are reported to have a greater frequency and size of RLS on TEE or transcranial Doppler (TCD) bubble studies.’(Page 5 Paragraph 3).
“The severity of RLS on TCD appeared to correlate with the Risk of Paradoxical Embolism score (RoPE) in one study,[13] suggesting it may be predictive of pathogenic PFO in contrast to incidental PFO, similarly to RoPE based on demographics and co-morbidities.” Please omit the statement after the last comma (similarly…). It does not make sense.
Thank you, we have amended it as per the reviewer’s suggestion.
“In this cohort study of patients with cryptogenic stroke and medically-managed PFO, we aim to investigate the association of shunt size with the risk of recurrent stroke, mortality and new AF, and identify predictors of recurrent stroke.” Please replace with ““In this cohort study of patients with cryptogenic stroke and medically-managed PFO, we aim to investigate the association of shunt size with the risk of recurrent stroke, mortality and newly detected AF, and to identify predictors of recurrent stroke.”
Thank you for the suggestion. We have rephrased it as per the reviewer’s suggestion.
Table 3. Please align the comorbidities section so as the numbers coincide with the correct line labels.
Thank you for the feedback. We have formatted the table accordingly.
We thank the reviewer for the detailed and insightful comments that have helped to strengthen the quality of the study findings.
Reviewer 2 Report
The paper by Lim et al. deals with an interesting topic. However, the more important weakness that this paper has is the fact that, as far as I understand, that they have about 300 patients on whom they run a multivariate analysis with an unspecified number of predictors. Authors correctly pointed it out in the limitations section. But the major flaw in this situation is that the number of events they have detected is very low (about 11 on 300 patients). In this case, I guess that multivariate approach would lead with high probability to an overfitting of results. Without an external validation I think that these results are not applicable to other populations.
Author Response
The paper by Lim et al. deals with an interesting topic. However, the more important weakness that this paper has is the fact that, as far as I understand, that they have about 300 patients on whom they run a multivariate analysis with an unspecified number of predictors. Authors correctly pointed it out in the limitations section. But the major flaw in this situation is that the number of events they have detected is very low (about 11 on 300 patients). In this case, I guess that multivariate approach would lead with high probability to an overfitting of results. Without an external validation I think that these results are not applicable to other populations.
We thank the reviewer for the encouraging feedback and the comment. In our multivariate regression model, we included three variables, namely shunt size, RoPE score, and LA diameter >40mm. We agree that in our cohort with 11 recurrent stroke events, a multivariate analysis has a high probability of overfitting. Hence, we have included this important point in our Limitations section. For convenient reference, the sentence reads, ‘In addition, the low number of recurrent stroke events increased the probability of overfitting the multivariable regression model, thus limiting the applicability of the results to other datasets.’ (Page 13 Paragraph 3)
Reviewer 3 Report
Though a monocentric study, the data is carefully analzyed and presented well. The message is straight forward and clear.
Author Response
Though a monocentric study, the data is carefully analzyed and presented well. The message is straight forward and clear.
We thank the reviewer for the encouraging feedback.
Round 2
Reviewer 1 Report
The authors have addressed adequately the issues that I raised in my previous review.
I have one last comment: Please include in the manuscript two important references which I used in my first review (1. Koutroulou et al. in Frontiers in Neurology 2020, doi: 10.3389/fneur.2020.00281. eCollection 2020 and 2. Koutroulou I et al. TAND 2020, doi: 10.1177/1756286420964673).
I congratulate the authors on this nice piece of work.
Author Response
We thank the reviewer for his kind comments.
Please see how revised manuscript with your recommend revisions.
1) Please include in the manuscript two important references which I used in my first review (1. Koutroulou et al. in Frontiers in Neurology 2020, doi: 10.3389/fneur.2020.00281. eCollection 2020 and 2. Koutroulou I et al. TAND 2020, doi: 10.1177/1756286420964673)
We have included the 2 references in the 'Introduction' paragraph 1, line 72: "However, with its high prevalence in the general population, PFO may be incidental in some cases of cryptogenic stroke.(5, 6)"
Thank you for your thorough review of our paper, and we thank you for the critical revisions.
Reviewer 2 Report
Authors replied to my comment adding an acknowledgement of this point in the discussion.
Author Response
Thank you for your kind review of our manuscript. We are grateful for the critical revisions provided to our manuscript.